# A Review on Microstructural Formations of Discontinuous Fiber-Reinforced Polymer Composites Prepared via Material Extrusion Additive Manufacturing: Fiber Orientation, Fiber Attrition, and Micro-Voids Distribution

**DOI:** 10.3390/polym14224941

**Published:** 2022-11-15

**Authors:** Zhaogui Wang, Zhenyu Fang, Zhongqi Xie, Douglas E. Smith

**Affiliations:** 1Department of Mechanical Engineering, Naval Architecture and Ocean Engineering College, Dalian Maritime University, Dalian 116026, China; 2Houston International Institute, Dalian Maritime University, Dalian 116026, China; 3Department of Mechanical Engineering, School of Engineering and Computer Science, Baylor University, Waco, TX 76798, USA

**Keywords:** DFRPC, MEAM, fiber orientation, fiber attrition, micro-voids

## Abstract

A discontinuous fiber-reinforced polymer composite (DFRPC) provides superior mechanical performances in material extrusion additive manufacturing (MEAM) parts, and thus promotes their implementations in engineering applications. However, the process-induced structural defects of DFRPCs increase the probability of pre-mature failures as the manufactured parts experience complicated external loads. In light of this, the meso-structures of the MEAM parts have been discussed previously, while systematic analyses reviewing the studies of the micro-structural formations of the composites are limited. This paper summarizes the current state-of-the-art in exploring the correlations between the MEAM processes and the associated micro-structures of the produced composites. Experimental studies and numerical analyses including fiber orientation, fiber attrition, and micro-voids are collected and discussed. Based on the review and parametric study results, it is considered that the theories and numerical characterizations on fiber length attrition and micro-porosities within the MEAM-produced composites are in high demand, which is a potential topic for further explorations.

## 1. Introduction

Material extrusion additive manufacturing (MEAM), otherwise known as Fused Filament Fabrication (FFF), or Fused Deposition Modeling (FDM^TM^ [1]) more commonly, offers the ability to rapidly build intricate structures at a low cost, and thus stands as a most popular manufacturing approach in modern automotive, aerospace, and other advanced industries [2]. Among several additive manufacturing techniques, MEAM stands out in terms of the range of applicable materials [3]. As the demand for lightweight, high-strength materials is continuously growing in aerospace and automotive industries, engineering applications of discontinuous fiber-reinforced polymer composites (DFRPCs) are becoming of interest. DFRPCs exhibit superior material stiffness and strength as compared to their virgin polymer alternatives [4]. They also reduce the thermal expansion behavior of the deposited material, and thus enhance the dimensional stability of MEAM-produced parts [5]. Meanwhile, DFRPCs can be more easily processed via MEAM systems as compared to their competitor, the continuous fiber-reinforced polymer composites (CFRPCs). To this end, DFRPCs are largely adopted in the recent-emerging large area extrusion deposition additive manufacturing (LAAM) technology as a convenient, low-cost, and efficient feedstock choice. LAAM is an MEAM-based approach, wherein a screw extruder is often involved with pelletized feedstock. A typical process of LAAM is to melt polymer composite pellets through the screw extruder, and to then deposit the molten materials onto a substrate in a relatively high flow rate (i.e., 2–10 kg/h for commercially available systems [6], and ~100 kg/h or higher for those mounted in research institutes, e.g., the super LAAM system in the University of Maine [7]). LAAM is extraordinarily useful in the rapid fabrication of parts and/or tooling in large-dimension (e.g., whole parts for full-size cars [8,9], naval applications [10,11,12,13], and large-dimension tooling [14,15]), as shown in Figure 1. In the particular COVID-19 context, it helps in rapidly building essential infrastructure for emergency medical purposes [16].

As shown above, due to the benefits of MEAM in rapidly fabricating lightweight structures with high stiffness and strength-to-weight ratios, we see continuously increased applications employing MEAM-produced DFRCPs as machine tooling or end-use engineering parts [8,10,11,12,17,18]. Nevertheless, a universally applicable set of processing parameters for MEAM (especially for large-scale MEAM) has not been found. This is mainly due to the fact that the complex thermal–mechanical physics occurring during the MEAM process cannot be controlled perfectly (cf. Figure 2). As mentioned, a typical process of MEAM implies the extrusion of molten thermoplastic feedstock through a nozzle orifice, which is then deposited on a pre-heated material platform. The viscoelastic nature of the thermoplastic-based materials makes the molten material flow, and the subsequent solidification, inter-beads wetting, and bonding make it hard to control from one type of polymer to the other [19,20,21]. A first and inevitable cause is that MEAM is inherently employed to create structures with complex geometries [22,23], and thus the deposited bead layer thickness [24,25,26,27], building orientation [28,29,30,31,32], print toolpath [33,34,35], infill structures [36,37,38,39], density [40,41,42,43], and so forth may drastically vary from part to part. Hence, the studies focused on the optimization of the print structure in the form of a multi-parameter combination with objective optimization effects, even with the aid of some advanced technology (i.e., the artificial neural network [44,45,46]), while they mostly could only offer qualitative guidance [47,48,49,50]. Due to the layer-by-layer fashion of manufacturing, MEAM structures inherently exhibit anisotropic mechanical responses subjected to different directions of external loads [51,52], e.g., lower bending strength and stiffness can be seen in the direction of layer accumulation as compared to those along in-plane deposition directions [53]. Gao et al. [54] suggested that the weak interlayer bond interfaces are a main contributor for the MEAM parts to exhibit weak and anisotropic mechanical properties. Another standpoint is that inter-filaments voids introduced in the additive manufacturing process greatly contribute to anisotropy [55]. The explanation is that the porous structures or voids can affect the mechanical performances of MEAM parts [56], which are prone to be undermined [57,58,59]. When operating as a load-bearing component, an MEAM part fails much easier when the external load direction is perpendicular to the material deposition plane, since severe stress concentrations are more likely to occur nearby meso-voids among interlayers in such conditions, as compared to those where load direction is parallel to the material deposition direction [60]. In addition, the bond formation mechanisms for amorphous and semi-crystalline polymers are different, which leads to a different degree of material anisotropy (e.g., PLA parts at around 50%, and PA which are less anisotropic, on an order of 10% [54]). This implies another important feature of polymers in contributing to the material’s anisotropy, which is the flow-induced polymer crystallization [61]. Brenken et al. predicted the non-isothermal crystallization for a semi-crystalline fiber-filled PPS and the non-uniform crystallization of an MEAM part was attributed to an anisotropic material performance factor [62]. Thermoplastics feedstock applied in FFF often exhibits intensive viscoelastic behaviors, e.g., the extrudate swell occurs during the extrusion deposition process [63], which greatly affects the print resolution and thus leads to the formation of inter-beads voids [60]. Therefore, the layer-by-layer-formed meso-structure is often considered as an inherent cause for the material anisotropy of MEAM-printed parts. In order to address such issue, the prior literature provided insights by explaining the meso-structural formations of MEAM-produced parts [53,64].

As discussed, the meso-structures of MEAM-produced thermoplastics are relatively weak, notably in the case of DFRPCs, wherein the fabricated structures exhibit more complex material properties owing to the inhomogeneous micro-structures formed by compounding discontinuous fibers and the polymer matrix [65]. As the DFRPC feedstock is extruded and deposited on a moving platform, velocity gradients within the melt orientate the suspended discontinuous fibers, and the final orientation pattern within the deposited beads directly affects the material properties of the solidified products. The narrow-gap shear-dominated flow in the nozzle die orifice induces a highly aligned fiber orientation along the direction of material loading [66], and thus leads to anisotropic material behaviors in macro-view. The prior literature tested the material properties of MEAM-made parts, wherein the deposited composites’ elastic moduli [67], thermal conductivity [68], and thermal expansion coefficient [5] were found to exhibit strongly anisotropic behaviors. Numerical studies were carried out in regarding the special behaviors of MEAM parts. Brenken et al. [69] performed a finite element simulation to study the thermal history of 50 wt.% CF/PPS in an MEAM process, wherein the anisotropic thermal conductivity of the short-fiber polymer composite was found. Compton et al. [70] simulated the time-dependent temperature contour of a LAAM-produced carbon fiber-filled ABS (CF/ABS) part through the finite element method, wherein an anisotropic thermal conductivity was assumed. Their solution implied that higher thermal conductivity is shown to be detrimental to the success of the build. Hoskins et al. [71] modelled the coefficient of thermal expansion of the deposited beads using a non-homogenized modeling approach, wherein the locally measured fiber orientation states were applied. The simulated results of the residual thermal stress within a printed cuboidal part of the CF/ABS were shown to be in line with the experimental scanning results [72]. Among various MEAM structures, the sandwich beam is one of the preferrable structures by virtue of its ability to combine the advantages of different materials, wherein the DFRPCs’ properties can be better exploited. Li and Wang [73] tested the sandwich composites wherein a carbon fiber-reinforced polymer (CFRP) was introduced as face sheets, as well as three types of inclusions. Compared to the conventional honeycomb-shaped inclusion, the sandwich beams embedded with a re-entrant honeycomb presented better energy absorption abilities. Hou et al. [74] compared three types of lattice composites inclusion, wherein CFRPs were used as face sheets, and wherein three kinds of core topologies were applicable for different impact-loading circumstances. Hassan et al. [75] proposed a finite element model to investigate the effect of different parameters of inclusion on the sandwich beams’ mechanical behavior. Essassi et al. [76] analyzed the fatigue behavior of the sandwich composites with four kinds of core densities, which possessed different stress ratios. It was found that a sandwich structure with a low core density can withstand a low maximum load, while its fatigue life is longer, which means that there must be a compromise between load bearing and fatigue life.

The previous literature has focused on the complexity of fiber-reinforced polymer composites (FRPCs) produced via MEAM systems, wherein the anisotropic material properties of MEAM-produced polymer parts [77,78,79,80] and the FRPCs [6,81,82,83] were mainly discussed. These studies provided valuable insights in explaining the effects of the MEAM building orientation and process parameters on forming the anisotropic material properties of polymers and their fiber-filled composites. In particular, we collected a few review articles within the last four years which summarize the employment of FRPCs in MEAM applications, including the small-scale (e.g., FFF, FDM) and large-scale (e.g., LAAM) systems, as shown in Table 1. Generally, the review discussions included the studies on the micro/meso/macro levels of the material performances, as well as the associated limitations generated by bringing in fiber fillers. Parandoush and Lin [84], Goh et al. [85], and Daminabo et al. [3] extended the focus of the polymer composites’ performances beyond MEAM, and other popular AM techniques were included. This provides a more extensive picture for readers to learn about polymer behaviors in AM processes. However, thermosets in SLS or SLA, e.g., exhibit distinctive different rheology properties as compared to thermoplastics that are normally employed in FFF or MEAM, and thus a broader introduction may mislead the reader who does not know every process in detail. Brenken et al. [6], Fallon et al. [86], and Papon and Haque [82] focused more specifically on the MEAM technique. Fallon et al. [86] suggested the importance of the melt viscosity of the filled polymers, which may yield more difficulty on the extrusion and deposition process as compared to virgin polymer alternatives (e.g., nozzle clogging). Brenken et al. [6] provided a summary on the tensile performances of polymer composites produced via MEAM, wherein discontinuous and continuous fiber fillers were included. Papon and Haque [82] updated the summary on the basis of Brenken’s work. They both focused on the micro/meso/macro levels of the material performances. However, the micro level discussions are not well exhausted in Papon’s work. Brenken did provide a good summary on the identification of material flow and associated fiber orientation in MEAM. In addition to that, within the last five years, MEAM is moving forward rapidly from small-scale to large-scale manufacturing, which brings in more opportunities and possibilities in engineering applications. MEAM products are no more than just prototyping, but they are employed as end-use parts and tooling in automotive, ship building, and other engineering fields. During this transition, some of the known knowledge of MEAM composites also changed, due in part to the variations in material feeding systems, processing parameters, and printing strategies. Nieto et al. [12] provided a review on large-scale MEAM, but it focused more on naval applications and little attention was given to the material itself. Recently, Mustapha et al. [87] provided a systematic review discussing the MEAM-produced polymeric composites that are applied to smart applications, e.g., 4D printing and so forth. This indicates a promising developing direction for MEAM and associated composites productions. Moreover, the molecular level, more than the microscopic level, can also have a significant impact on the MEAM composites. Specifically, molecular orientation is important for the mechanical performance of short fibers [88,89,90] as reinforcements. In the papers, the functionalities of the produced composite were the main focus, while the technique theory of the composite’s microstructure and related material performance, or the more microscopic orientation of the molecules inside the material, were not well presented.

It is appreciated that prior reviews summarized the remarkable developments of MEAM-produced composites in engineering applications. Despite these advances, we note that there is a lack of an in-depth investigation on the micro-structural formations of MEAM-produced DFRPCs. From a technical perspective, we believe a more comprehensive discussion on research articles exploring the microstructures of DFRPCs can bridge the knowledge gap between MEAM processing parameters and a preferable composite product with superior mechanical properties. To this end, we continue to investigate the microstructural formation of the DFRPCs produced via MEAM (cf. Figure 3). The following sections are planned as follows: Chapter Two introduces the flow-induced fiber orientation and collects the related literature focusing on MEAM applications. Chapter Three discusses the fiber length attrition during the MEAM process and compares the fiber length studies in a few screw-based extrusion processes. Chapter Four presents the micro-voids formed due to the fiber-related features (e.g., fiber orientation). The last chapter summarizes the experimental and numerical achievements obtained by the current literature. In order to obtain superior DFRPC products with preferable micro-structures via MEAM systems, we finally project the urgent topics to be further explored in the context of MEAM–DFRPCs’ microstructural formations.

## 2. Fiber Orientation

During the process of MEAM, discontinuous fibers re-orientate in the material flow as they pass through the extrusion die, induced by the strong shear-dominated and stretching flow formed in the narrow gap orifice. Tekinalp et al. [66] found that ~90% of short fibers were aligned along the direction of material flow through the experiments, where a desktop-size FFF 3D printer (nozzle die diameter: 0.4 mm) was adopted to test the short carbon fiber-filled ABS polymer. Mulholland et al. [91] measured the fiber orientation of the copper fiber-filled polyamide 6 composites prepared through a twin screw extruder with a 1 mm nozzle die [55]. The alignment along the flow-direction was ~80% [91]. The method of ellipses (MoE) is a commonly employed method to quantify the fiber orientation state of a polymeric composite part. Nargis [92] prepared a LAAM-deposited 13 wt.% CF/ABS polymer part and polished the cross-section of the sample. Through microscopic imaging, the complete or incomplete elliptical and rectangular footprints of the fibers can be obtained digitally. For cylindrical fibers, at the cross-section which appears on the intercepting plane, an elliptical image can be observed. The characteristic values of these micrographs can be measured in the computer. Subsequently, the fiber orientation tensors of the sampled spot, representing the fiber orientation state of the imaging area, can be calculated. Accordingly, the LAAM-deposited beads exhibited a lower flow-direction fiber alignment expectation than those produced by the desktop-size MEAM 3D printers. The A_33_ of the measurement orientation tensors (i.e., indicating the flow-directional fiber alignment) is ~0.5–~0.6, while the results from Tekinalp et al. [66] are as high as 0.92. The comparison implies that the feeding mechanisms of the small-scale and large-scale MEAM systems can be a vital factor in forming the material properties of the produced composites. In addition, it should be noted that the MoE can be easily applied with a fine polisher and relatively low-cost microscopic instruments, as shown in Figure 4. Nevertheless, the sampling areas are often limited and the experimental procedures of the MoE can be tedious.

In contrast, micro-CT scans are employed for rapidly understanding the microstructural formations of MEAM-produced composite parts. With the advanced non-destructive, high-resolution X-ray micro-analyzers, one can easily detect the morphology of a composite part through a long distance. Yu et al. [93] measured the fiber orientation of BF/PLA composites (filled with different contents of basalt fibers) prepared via FFF 3D printing and mold processing, respectively. The computed orientation tensor suggests that the flow-directional orientation of the FFF-prepared parts are ~70% higher than those molded (i.e., ~0.81 versus ~0.47). Yang et al. [94] used the micro-CT approach to quantify the fiber orientation of the material flow in an FFF nozzle. They found that the fiber orientation state parallel to the principal material-loading direction significantly reduces from ~0.9 to ~0.7 during the extrusion–deposition transition, as shown in Figure 5. Additionally, Somireddy et al. [95], Sommacal et al. [96], Hmeidat et al. [97], Chisena et al. [98], Tagscherer et al. [99], Yeole et al. [100], and Kumar et al. [101] also employed the micro-CT scanning method to obtain an in-depth exploration of the MEAM-produced composites’ microstructures, which helped explain the associated mechanical performances of the composites. It is clearly seen that the micro-CT approach is a convenient and powerful tool to explore the internal micro-structural formations of the MEAM-produced composites, and in a non-destructive way as well. We also note that the micro-CT method is usually costlier than the traditional MoE approach.

As seen, a fiber orientation tensor is extensively applied into the quantitative evaluation of the fiber orientation state of the MEAM–DFRPCs.

Jeffery laid the groundwork for the fiber orientation tensor methods for polymer processing applications when he first deduced the motion of a single rigid massless ellipsoid in a purely viscous fluid [102]. According to Jeffery’s research, the unit vector p, which coincides with the longitudinal axis of the suspended stiff particle, determines the direction of a fiber (cf. Figure 6). A suspended ellipsoid’s equation of motion is given as [103]:(1)$$\dot{p}=W\cdot p+\gamma_{e}\left(D\cdot p+D:ppp\right)$$
where the symmetric and antisymmetric components of the flow’s velocity gradient, D and W, respectively, are the rate-of-deformation tensor and vorticity tensor:(2)$$D=\frac{1}{2}\left[\left(\nabla v\right)+{\left(\nabla v\right)}^{T}\right],\;\mathrm{and}\;W=\frac{1}{2}\left[\left(\nabla v\right)-{\left(\nabla v\right)}^{T}\right]$$

The equation of motion for the second-order orientation tensor A was additionally developed by Advani and Tucker [104] as:(3)$$\frac{DA}{Dt}=\left(A\cdot W-W\cdot A\right)+\gamma_{e}\left(D\cdot A+A\cdot D-2A:D\right)+2C_{I}\dot{\gamma }\left(I-3A\right)$$

Here, the material derivative is donated by $\frac{D}{Dt}.$ In Jeffery’s work, when the rotary diffusion is proportional to the scalar magnitude of D, appearing as $\dot{\gamma }$ in Equation (3), the final term enforces an isotropic rotary diffusion that excludes the tumbling motion. One phenomenological variable that can be altered to comply with the interaction between suspended fibers is the fiber–fiber interaction coefficient $C_{I}.$ Under large strains, the assumption of a steady orientation state is established by the Folgar and Tucker-proposed isotropic rotary diffusion (IRD) term [103]. The Advani–Tucker IRD model (or IRD model) is a common name for the model in Equation (3).

In contrast to the Folgar–Tucker model, Advani and Tucker [104] established fiber orientation tensors in order to quantify the orientation state for concentrated suspensions with significantly fewer independent variables [104]. The moments of the fiber orientation distribution function ψ(p) define a second-order orientation tensor $A=A_{ij}=\oint_{S}p_{i}p_{j}\psi \left(p\right)d\;S,$ and a fourth-order orientation tensor $A=A_{ijkl}=\oint_{S}p_{i}p_{j}p_{k}p_{l}\psi \left(p\right)d\;S$, wherein their method has found broad use in polymer composite molding [104]. It is important to note that the integrations are carried out over the surface S of the unit sphere, and the probability density function ψ(p) integral over the entire sphere is equal to unity, which constitutes a normalization requirement in A. This leads to the results that the trace of A is one and symmetric, for which the number of independent components in A is thus reduced to five [105]. As a consequence, the second-order fiber orientation tensor can be expressed compactly as $A=A_{ij}=\left[A_{11},A_{12},A_{13},A_{22},A_{23}\right]$, which offers a practical method for calculating fiber orientation in polymer melt flows. A variety of closure approximations, such as the hybrid closure [106], the natural closure [107], the invariant-based fitted closure [108], and the orthotropic closure [109], have been put forth. They are frequently used to approximate the fourth-order fiber orientation tensor A as a function of A. We note that MEAM-related flow/orientation studies are seen to use the orthotropic closure frequently [63,110,111,112]. It is also crucial to notice that the orientation tensor method does not track each single fiber, but rather indicates the degree of alignment through the nine components of tensor A. The second moment of the orientation distribution function δ(θ,ϕ) is represented by the second-order orientation tensor A. In particular, Figure 7 provides two vital examples of A: the case of a uniformly random orientation is represented by three diagonal components of A with equal values, namely 1/3; and a diagonal component of A with a value of one denotes an orientation fully corresponding to the corresponding direction.

In order to further explain the correlations between the MEAM material flow and the induced fiber orientation, numerical studies are carried out. Fiber orientation distributions in FFF nozzles with three distinct internal geometries have been calculated in the work by Nixon et al. by virtue of the Moldflow program (Moldflow Corporation, Framingham, MA, USA). The ultimate findings demonstrated that a higher fiber alignment was obtained by convergent nozzle geometry rather than by divergent nozzle shape [113], while the extrudate flow outside the nozzle was not evaluated. Heller et al. [63] employed a highly viscous Newtonian flow model to model a desktop-size FFF extrusion scenario, wherein the vertical extrudate swell physics was included. The high fiber alignment that occurs along the main flow direction was quantified and the die swell reduced the principal direction fiber alignment by ~20%, as the results show [63]. Wang and Smith extended Heller’s work by evaluating the effects of assumed polymer rheology [112], and swirling kinematics of the large-scale extruder feeding mechanisms [114] on the predicted fiber orientation of an LAAM nozzle flow and vertical free extrudate. Right after the molten polymer composites were extruded out of the die, the direction of the flow rapidly transited from vertical extrusion toward horizontal deposition. Additional shear forces applied to the flow make the fiber orientation within the flow more complex. Heller et al. simulated the polymer deposition process of a CF/ABS composite bead using a 2D planar flow model, and the results show that the flow region contacting the material substrate exhibited a different fiber orientation as compared to the upper region of the flow, as shown in Figure 8 [110]. Russell et al. employed the planar flow model used by Heller et al. [110] to compute the effective thermal expansion coefficient of an LAAM-made composite bead using the orientation homogenization method [111], and the numerical predicted elastic constants of a 13 wt.% CF/ABS were in line with the related experimental work [67]. Nevertheless, the above literature studied the flow and fiber orientation under a one-way weakly coupled formulation, wherein the flow kinematics were computed by ignoring the presence of fibers. In fact, the fiber orientation alters the rheological behaviors of the material flow, and thus an interactive coupling relationship is found between the flow and the fiber orientation. This will in turn affect the mechanical and thermal properties of the deposited beads and is thus of great importance. Mezi et al. modelled a fully coupled Newtonian fiber suspension flow for the FFF extrusion process, wherein a modified Tanner model was applied to capture the die swell of fiber composite melt flow [115]. Bertevas et al. [116], Yang et al. [117], and Ouyang et al. [118,119] simulated the flow–fiber orientation coupling behavior in the bead deposition process of fiber-reinforced composites using the smoothed particle hydrodynamics (SPHs) approach, wherein the dynamic fiber orientation’s evolutionary process during the extrusion deposition transition was evaluated (cf. Figure 9). Alternatively, Wang and Smith employed a finite-element-based algorithm to evaluate the mutually dependent effect between the polymer flow rheology and the fiber orientation in the MEAM nozzle–extrudate flow [120] and the 2D planar extrusion–deposition flow [121], wherein a quasi-steady state of fiber orientation in the deposited composite was computed for further material properties estimations. Recently, Wang employed the advanced pARD-RSC fiber orientation prediction model into the fully coupled flow/orientation analysis algorithm [122]. As shown in Figure 10, the predicted fiber orientation results, with an assumed 3D random fiber orientation initial condition at the nozzle inlet, exhibited a favorable agreement with a related experimental report [92]. It is generally acknowledged that the fiber orientation formation of the deposited beads directly affects the macro-material anisotropic behavior of the produced composites. Current studies are mainly limited by 2D simplified flow models, and thus the predicted fiber orientation tensor results cannot present the material anisotropy of the entire deposited beads. We note that the numerical studies on the 3D deposition flow modeling of MEAM processes have been presented (e.g., [123,124,125]), and thus the flow/orientation analyses based on these 3D flow models are expected in future studies.

## 3. Fiber Length Attrition

Fiber length is one of the most crucial factors that determines the properties’ enhancement of reinforced polymers. In desktop-size MEAM processes (e.g., FFF), the fiber length stays stable from the filament feedstock to the deposited parts. As shown from Jiang and Smith [126] (cf. Figure 11), we see that the fiber length distributions in filament and printed conditions exhibit subtle differences for all examined filled polymers (including ABS, PLA, PETG, and Amphora^TM^). Although the FFF extrusion and deposition processes yield little fiber length attrition, the fraction of filled fibers plays an important factor contributing to fiber length attritions. Tekinalp et al. [66] measured the fiber length distributions of CF/ABS with different weight fractions prepared via compression molding (CM) and FDM. As the results show in Figure 12, it is seen that the averaged fiber length values reduced significantly with an increased fiber weight fraction, wherein the trend of reduction in the FDM samples was much higher than that of the CM samples. This indicated that the fraction of filled polymer feedstock applied in FDM (or say, filament-based MEAM) may have had an effective fraction limitation, i.e., a higher filled fraction may suffer significant fiber length attrition and result in under-expected property improvement in the produced composites. This phenomenon was also reported by Ning et al., wherein the tensile properties and associated micro-structural fiber length attritions of FDM-produced CF/ABS samples were studied [127]. Therefore, we can see that the commercially available fiber composites for filament-based MEAM systems are often filled with ~10 wt.% fiber contents, balancing the cost and effectiveness of property enhancements.

We note that the nozzle diameter of a desktop MEAM system is often 400 μm, and thus it is not surprising that longer fibers would not survive through the extrusion deposition process to the final printed parts. Nevertheless, the large-scale MEAM systems with screw-extruders employed are often equipped with nozzles in much larger diameters (e.g., in the magnitude of ~1 mm, or even ~10 mm). The upgraded extrusion die allows for the longer fibers to survive, and then a higher increment in material properties in the deposited composites is expected. Nevertheless, the revolute material feeding system may yield considerable geometry loss for the reinforced discontinuous fibers. Berzin et al. performed fiber geometry measurements along the flow direction of a twin-screw extruder (cf. Figure 13 and Figure 14), and both the length and diameter exhibited more than 100% reductions at the end of the screw, as compared to their starting dimensions [128]. Hausnerova et al. showed that the high shear stress generated during the screw rotation has a direct effect on reducing the length of the reinforced fibers, and the composites with increasing fiber volume fractions exhibited a decreasing averaged fiber length after screw processing [129]. Aigner et al. employed the X-ray computed tomography approach in measuring the fiber breakage of a glass fiber polymer composite processed through a single screw extruder, wherein the maximum fiber length of the extruded composites was reduced by ~50%, as compared to data provided by the manufacturer [130]. Goris found that the melting temperature was also attributed to the degradation of fiber length of long fiber-reinforced composites in an injection molding application [131]. Similar studies carried out by Zhuang et al. [132] and Bailey and Kraft [133] indicated that the processing parameters’ residence time and molding pressure were contributors to fiber length loss. Bayush et al. measured the fiber length distribution of screw-extruder-processed hemp fiber-reinforced polypropylenes and the mechanical and dynamic properties of the compound were tested. The results indicated that maintaining a relatively larger averaged fiber length benefited the mechanical performances of the natural fiber composites [134]. Gamon et al. [135] suggested that maintaining the fiber length in higher values enhanced the flexural behavior of screw-extruded composites. Inoue et al. [136] studied the effect of the screw design on the fiber breakage and dispersion and reported that the fiber length was a direct factor in determining the mechanical properties of the mixed composite. For this reason, we should not be surprised that discontinuous fiber composites experience fiber geometry loss during LAAM.

Duty et al. employed two different designs of the screw barrel in an LAAM system and elastic modulus along the print direction of the deposited 20 wt.% SGF/ABS (i.e., short glass fiber-filled ABS) exhibited a difference of 42% [67], implying that the screw design had an influence on the fiber length attrition and associated elastic properties of the deposited beads. Russell and Jack [137] and Wang et al. [138] separately measured the fiber length distributions of pellets and deposited beads of 13 wt.% CF/ABS that were used in LAAM applications. They continued by employing the fiber length distribution data in a homogenization approach for evaluating the elastic properties of short fiber composites, and the results suggested that the longer fibers led to higher tensile moduli [137,138]. Yeole et al. measured the fiber length distribution of 50 wt.% CF/PPS processed via a large-scale MEAM system [100], wherein a 10.16 mm-nozzle (diameter) was employed. The measured data of pellet feedstock and deposited beads are presented in Figure 15, where the fibers exhibited little length reduction in comparing the data between the pellets and the beads. The averaged fiber length, on the other hand, was above ~300 μm, which was higher than what we have normally seen in desktop-size systems produced (i.e., 50–100 μm). In addition, the fiber fraction of the feedstock reached 50 wt.%, which was also higher than FDM-used feedstock (e.g., Jiang ang Smith [126], Ning et al. [127]). The higher fiber length and fiber fraction obtained by the large-scale system indicate that large-scale MEAM systems are promising in producing composite structures with superior mechanical properties. Consequently, we need to note that higher length of fibers also reinforces the material anisotropy of MEAM-produced composites, and thus a trade-off may be needed in selecting proper filled polymer feedstock based on different application demands.

In order to further understand the fiber length attrition behaviors, theoretical and numerical studies were conducted. Bereaux et al. [139] modelled the bending moments of a single fiber as it passed by the screw-generated shear flow, and the fiber length distribution resulted from the extruded composites was computed. They stated that the fiber fracture occurred when screw-applied shear stress exceeded the critical bending fiber length, which indicated the importance of the screw’s design in retaining the fiber length [139]. In the injection molding process of long fiber thermoplastics (LFTs), Phelps et al. [140] modeled the fiber breakage phenomenon by considering the fiber buckling effects. They correlated the fiber length attrition with the fiber orientation state, as shown in Figure 16, wherein the predicted results obtained by Phelps showed a good agreement with the corresponding experiments on glass-fiber/polypropylene LFTs molding [140]. Bechara et al. [141] recently presented a phenomenological constitutive model for the fiber breakage modeling of LFT molding parts based on rheological experiments on simple shear flows of the glass fiber-filled PP polymer (cf. Figure 17). The model was based on the beam theory, wherein the fiber–fiber interactions were considered together with the fiber volume fraction via a fitting parameter (i.e., phenomenological parameter). This model can be used to track the number-average and weight-average fiber length values during the injection molding process, which is practically useful. Nevertheless, the numerical models are limited to molding process applications. The MEAM extrusion deposition process takes place mainly under lower pressure and temperature conditions, as compared to typical molding processes, and thus the fiber buckling and associated breakage may exhibit differences. We note that it is important to develop constitutive models to depict the fiber length reduction in MEAM applications, especially for the large-scale systems.

## 4. Micro-Voids

While fiber reinforcements are expected to improve the material properties of MEAM-deposited composites as compared to the alternative virgin polymers, the MEAM-flow-induced variations of the fiber-related micro6structures (e.g., fiber orientation, fiber attrition, fiber migration, and/or imperfect fiber/matrix bonding) can result in a significant amount of micro-porosities within the produced composites. Al-Maharma et al. [142] conducted a critical review on the correlations between the micro-voids in the additively manufactured parts and their macro mechanical properties. These defects can potentially impact the fatigue strength, stiffness, mechanical strength, fracture toughness properties, and even corrosion resistance. Additionally, the existing micro-voids sometimes tend to concentrate the interfacial stresses, influencing the interlaminar bonding quality, resulting in the interfacial flaws, e.g., interfacial dislocation and delamination. Yu et al. [143] explored the contribution of printing-induced fiber alignment and voids coupled with the matrix to the anisotropic elastic modulus, assisted by computational tomography (CT). The relationship implied that the Young’s modulus, shear modulus, and Poisson’s ratio can be tailored by programming the printing direction. Kong et al. [144] studied the interfacial failure under the pure and mixed modes. The research showed that the interlaminar characteristics of different materials and fiber angles could be markedly different. Papon et al. [145] focused on overcoming such drawbacks by means of acid-based oxidation treatment and vacuum annealing, which remarkably increased crystallinity (~100%) and enhanced the fiber–matrix interfacial bonding. To this end, we consider the micro-void as a vital component to better understand the material properties of MEAM-produced composites. Chisena et al. [98] evaluated the fractions of porosity, fibers, and polymer matrix in MEAM-produced nylon composites via micro-CT scanning. They found that the printed beads near the heat-bed exhibited ×1.5 larger pores (i.e., >250 μm^2^) than those in the upper region of the printed sample (i.e., <100 μm^2^), which was considered as a result of the vertically formed large thermal gradients along the deposited beads (cf. Figure 18). Yeole et al. [100] explored the microstructural formations of 50 wt.% CF/PPS processed via molding processes and MEAM process. They found that the micro-voids of MEAM-printed beads were larger than those prepared via compression molding, as shown in Figure 19. Additionally, the highly aligned flow-direction fiber orientation was attributed to the generation of the micro porosities in the deposited beads. Somireddy et al. stated that the thicker layers also yielded more micro-voids in the deposited beads and thus led to more anisotropy in SCF parts [95]. Sommacal et al. [96] studied the micro-voids and related fiber alignment in filament-MEAM-produced PEEK material samples via micro-CT imaging. The results indicated that both the filament feedstock and the printed bead contained a significant amount of voids, and the printing process did not remove the voids originally presented in the filament. Kumar et al. [101] combined the large-scale MEAM and compression molding (CM) processes to reduce the micro-voids within raw MEAM-produced composites. Their results indicated that the combined AM–CM method effectively reduced ~50% of porosity volume fraction as compared to the raw MEAM-produced composites (cf. Figure 20). Tagscherer et al. [99] performed a fundamental study on the microstructural formation of large-scale MEAM-produced composite parts via X-CT imaging. Their results suggested that a higher layer thickness helped decreased the chance of micro-voids, and these micro pores were found in the core of a deposited bead rather than at the free surface boundary of the deposition flow.

Thanks to the rapid development of the non-destructive tomographic methods (e.g., micro-CT scanning), we see studies analyzing the fiber-related micro-voids formulations in MEAM processes, including filament-based and pellet-based (i.e., large-scale system). On the contrary, the literature addressing the micro-voids’ formation from numerical perspectives is scarce. Awenlimobor et al. explained the micro-void formation of the MEAM-produced composites by a finite element fiber suspension analysis [146], wherein the velocity gradients and the pressure distribution of a single fiber along a deposition flow streamline were presented. The pressure distribution of the fibers was considered as a direct factor determining the micro-voids’ formation near the fibers (cf. Figure 21). They also plan to continue the study with 3D flow modeling and correlate the micro-voids with flow shear rate and other kinematics information. As our review has shown, the numerical models and simulations depicting how micro-voids are formed during the MEAM process are still lacking. Nevertheless, the above experimental observations (e.g., [95,98,142]) suggest that the micro-voids within MEAM-produced composites yield significant impacts in reducing the mechanical performance of printed structures. Therefore, we see a high demand for numerical modeling works of micro-porosity analysis in MEAM flow studies.

## 5. Conclusions

MEAM (including the desktop-size FFF and large-scale LAAM systems) has been proved as a cost-effective approach for DFRPCs manufacturing. Within recent decades, MEAM-produced DFRPCs have seen continuous applications in aerospace, automotive, and naval industries. Nevertheless, the DFRPCs exhibit much more complex and non-homogeneous microstructural formations as compared to their virgin alternatives after being processed through an MEAM system. The flow-induced fiber orientation stands for a first and foremost factor attributed to the non-isotropic material behaviors of MEAM-produced composites. As the reviewed articles have shown, the short fibers of MEAM-printed beads highly align along the material-loading direction, and thus lead to a principal direction for the anisotropic material properties. The length of the reinforcements also plays an important factor in the enhancement of the DFRPCs’ material properties. The filament-based MEAM-composites exhibit little fiber length attrition by comparing the fiber length distributions of the filament feedstock and the deposited beads composites. In contrast, the LAAM systems have a screw-based material feeding mechanism that caused severe fiber breakage during the manufacturing, thus resulting in an uneven fiber aspect ratio distribution, which also adds complexity to the material anisotropy of the FFF-printed composites. Related studies have shown that the averaged fiber length of the deposited bead was much less than that of the pellets. We have seen that numerical models correlated the fiber breakage with the buckling of the fibers, which could be applied to further depicting the anisotropic properties of the LAAM-produced DFRPCs. Nevertheless, current theories are limited to long fiber composites produced via molding processes, which present notable differences from the MEAM process. Micro-voids in the deposited DFRCPs also crucially influence the mechanical performances of MEAM-produced composites. X-ray tomographic results also indicated that more voids were seen in thicker deposited beads, which was also attributed to the highly aligned fiber orientation (i.e., the fiber orientation within the deposited beads was found to highly align along the direction of material loading). In addition, the micro-porosity also contributed to the anisotropic material behaviors of the MEAM-produced composites.

Finally, we provide a summary of the obtained knowledge from the reviewed studies in Table 2. In the same table, we also project a few vital components that are needed for further characterizing the microstructural formations of MEAM–DFRPCs. Although significant improvements are achieved to bridge the knowledge gap between the microstructure formation of MEAM–DFRPCs and the process itself, we see a lack in the numerical studies for further exploring the correlations between the MEAM material flow kinematics and the micro-structural formation of the composite beads, e.g., 3D flow modeling of the extrusion–deposition process and associated analysis on the fiber orientation and molecular orientation, which is not well presented yet. In addition, the flow/orientation coupled analysis in 3D flows computation is expected as a highly non-linear problem, which is significantly challenging yet important in correctly identifying the rheology behaviors of the material flow. Furthermore, the shear-dominant extension flow of the LAAM systems also contributes to fiber attrition. Nevertheless, numerical models of fiber attritions associated with the extrusion–deposition processes have not been found, to the best of our knowledge. Moreover, there is still a lack of modeling techniques addressing the fiber/matrix bonding behaviors during the extrusion–deposition process, which is a major reason for the formation of micro-voids within deposited composite beads. The pressure distribution around the fibers can be a possible factor determining such issue, while more in-depth studies are still expected. Additionally, it is seen from a few of the reviewed experimental reports that the fiber orientation, fiber length attrition, and the fiber/matrix bonding related to void formation are somehow correlated with each other. However, coupled numerical studies on these factors are limited, partially due to the large computational costs of these coupled analyses. To this end, another crucial direction on micro-structural formation characterizations is the cost-effectiveness of the coupled numerical algorithms, e.g., parallelization computation on identifying the micro-structure factors may be needed. Meanwhile, proper assumptions are needed if decompositions were applied in coupled analyses (e.g., see [120,121]).

## Figures and Tables

**Figure 1 polymers-14-04941-f001:**
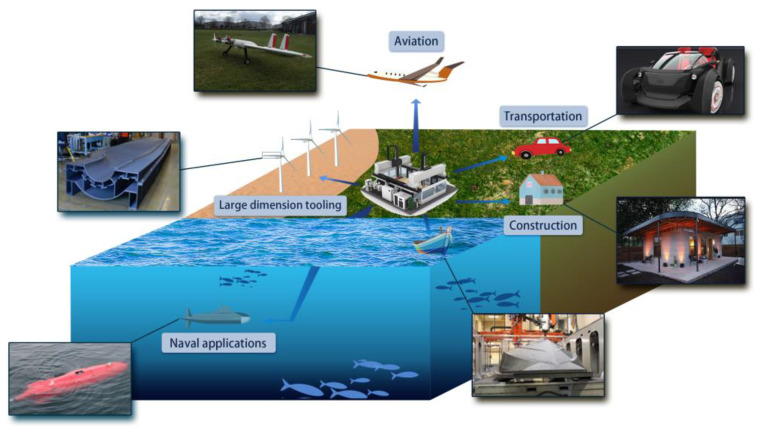
Widespread LAAM-produced composites’ applications in different engineering fields.

**Figure 2 polymers-14-04941-f002:**
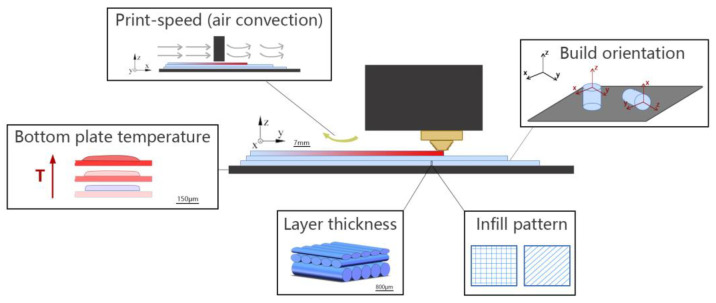
MEAM printing parameters that lead to weak meso-structure.

**Figure 3 polymers-14-04941-f003:**
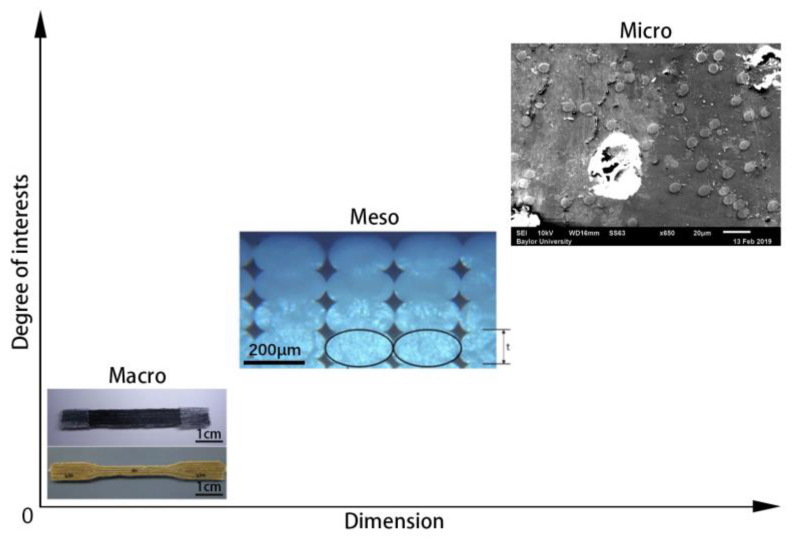
Structural formations of composite materials produced via MEAM.

**Figure 4 polymers-14-04941-f004:**
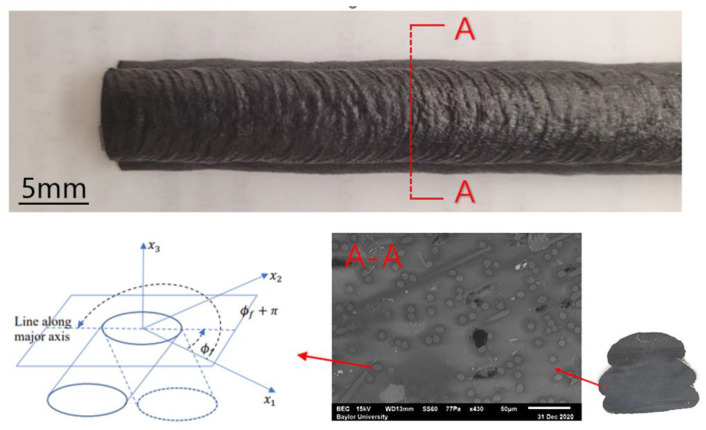
Fiber orientation measurement on an LAAM-deposited bead with MoE [92].

**Figure 5 polymers-14-04941-f005:**
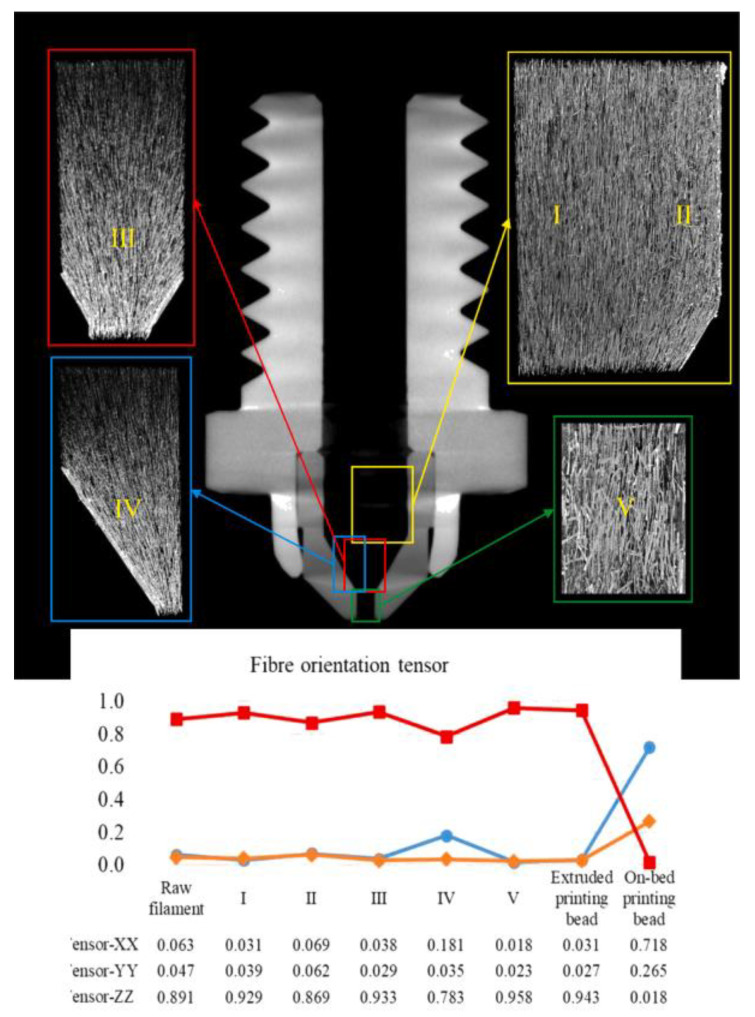
Fiber orientation tensor evaluation along an FFF extrusion nozzle [94].

**Figure 6 polymers-14-04941-f006:**
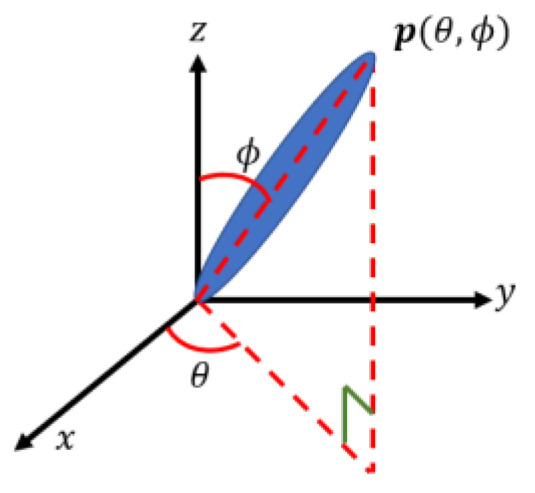
Vector representation of a single fiber’s orientation.

**Figure 7 polymers-14-04941-f007:**
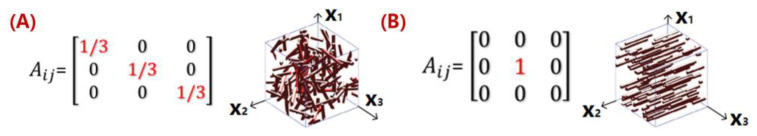
Fiber orientation tensors for (**A**) uniformly random alignment and (**B**) fully aligned along $x_{2}$ direction.

**Figure 8 polymers-14-04941-f008:**
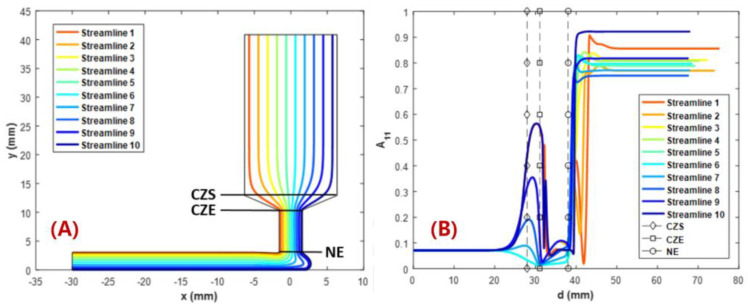
(**A**) Ten Evenly Spaced Streamlines for Three Flow Regimes.; (**B**) Fiber Alignment in the x Direction for Level Flow, A_11_ [110].

**Figure 9 polymers-14-04941-f009:**
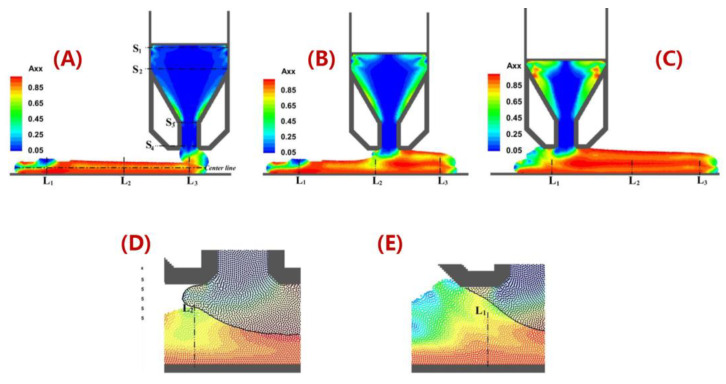
Instantaneous contour plots of the orientation tensor component Axx for the two-layer printing: (**A**) t = 0.12 s, (**B**) t = 0.15 s, and (**C**) t = 0.195 s. Here, (**D**,**E**) correspond to the enlargement of (**B**,**C**) in the regions L_1_ and L_2_, respectively, where both layers are highlighted (Φ = 0.03, a = 30) [119].

**Figure 10 polymers-14-04941-f010:**
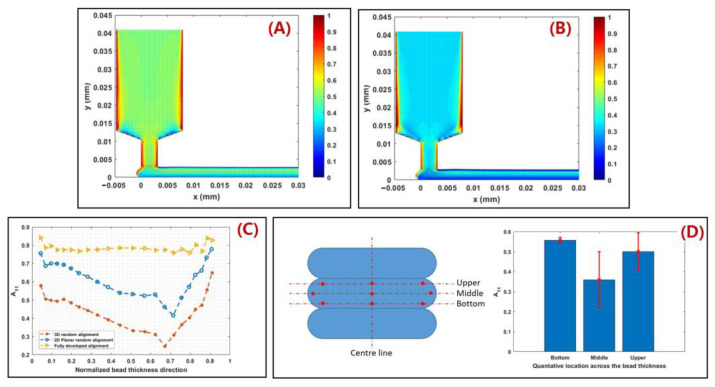
(**A**) Fiber orientation tensor A_33_ component contour of 2D planar deposition flow with assumed 3D random fiber alignment initial condition at nozzle inlet; (**B**) Fiber orientation tensor A_33_ component contour of 2D planar deposition flow with assumed 2D planar random fiber alignment initial condition at nozzle inlet; (**C**) A_33_ component at flow end with different assumed initial fiber orientation conditions; (**D**) Measured fiber orientation tensor component A_33_ [122].

**Figure 11 polymers-14-04941-f011:**
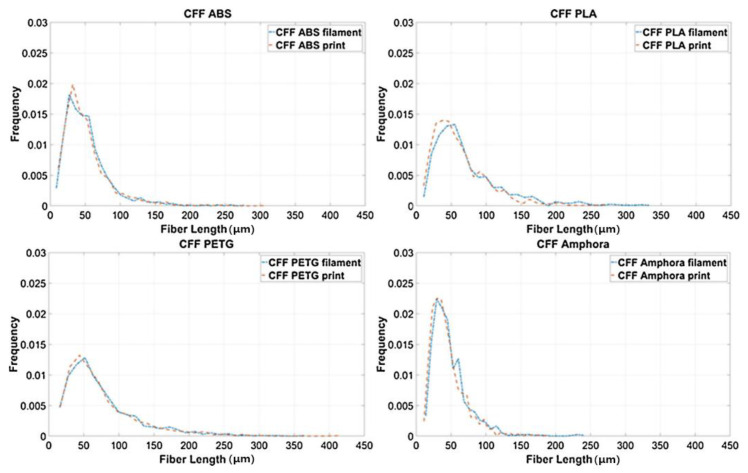
Fiber length distributions plots for each CFF material filament [126].

**Figure 12 polymers-14-04941-f012:**
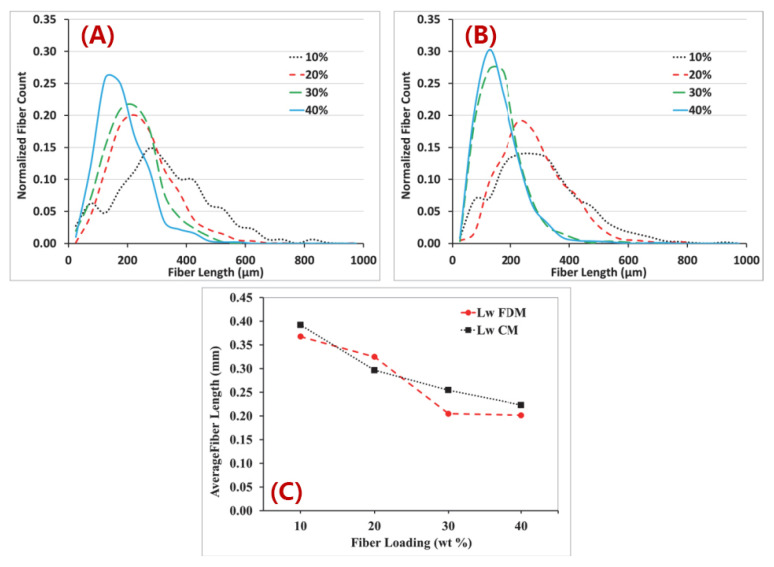
Fiber length distributions (**A**) compression-molded, (**B**) FDM-printed, and (**C**) weight average fiber lengths of dog bone samples [66].

**Figure 13 polymers-14-04941-f013:**
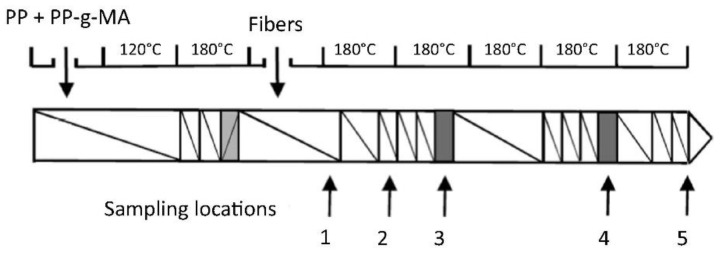
Scheme of the laboratory scale twin-screw extruder (Clextral BC21). Restrictive zones are in grey. Arrows indicate sampling locations [128].

**Figure 14 polymers-14-04941-f014:**
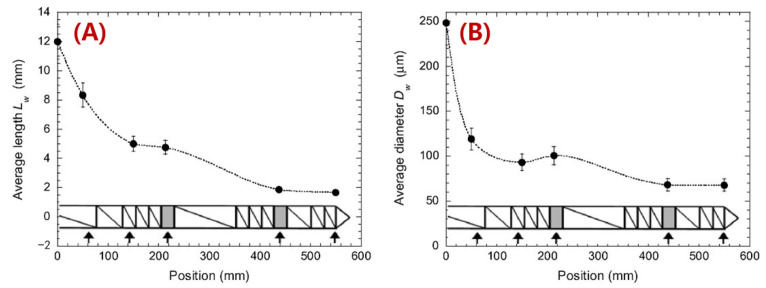
Changes in length (**A**) and diameter (**B**) of 12 mm flax fibers along the screws (2 kg/h, 100 rpm). Lines are just to guide the eyes [128].

**Figure 15 polymers-14-04941-f015:**
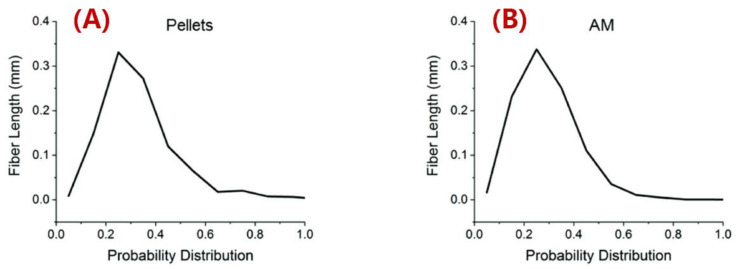
Probability distribution of fiber length: (**A**) pellets; (**B**) deposited beads [100].

**Figure 16 polymers-14-04941-f016:**
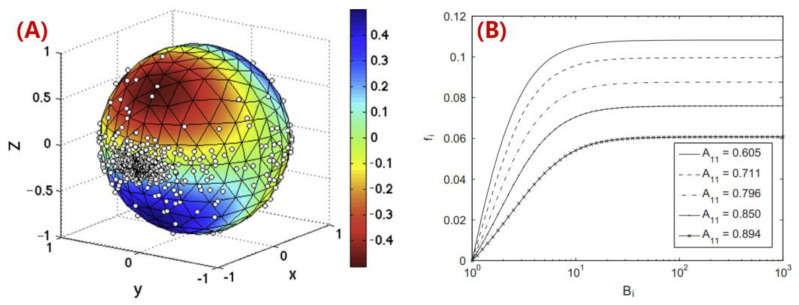
(**A**) Sphere of all possible fiber directions **p**, colored by the value of (**D**: **pp**) for the simple shear flow $v_{x}=\dot{\gamma }z$. Negative values (red to yellow colors) indicate orientations where the fiber is in compression. Points on the sphere are a sample of fiber orientations at steady state for this flow, calculated using the Folgar–Tucker model; (**B**) Fraction fi of fibers that have an orientation in which they can buckle, as a function of the buckling parameter B_i_, for various steady-state orientations in simple shear flow.

**Figure 17 polymers-14-04941-f017:**
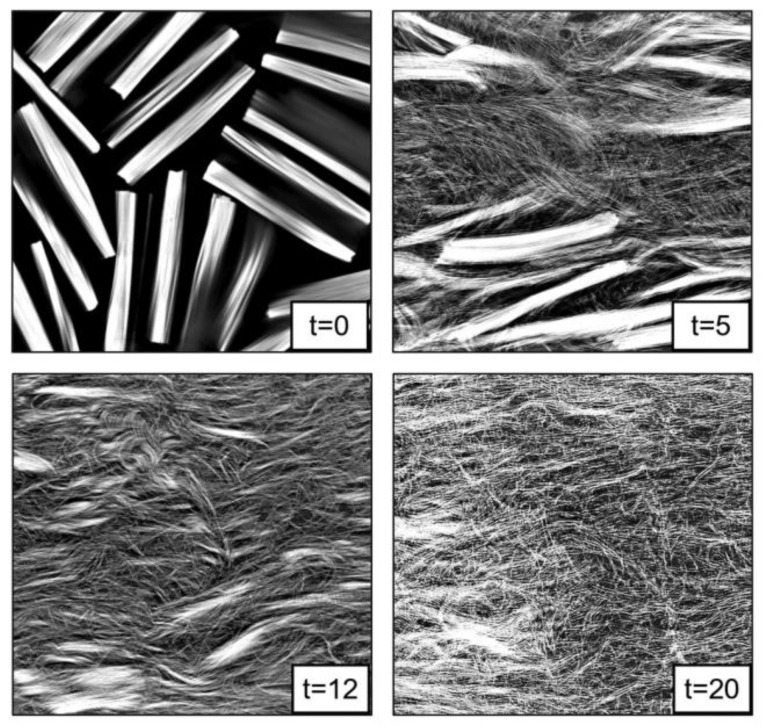
Micro-CT slices of fiber dispersion for PPGF40 (40 wt.% glass fiber-filled PP polymer) exposed to a simple shear flow at 50 s^−1^ for different residence times. Of note, the fiber diameter is 14–24 μm, and the density of fibers is 2.56 g/cm^3^ [141].

**Figure 18 polymers-14-04941-f018:**
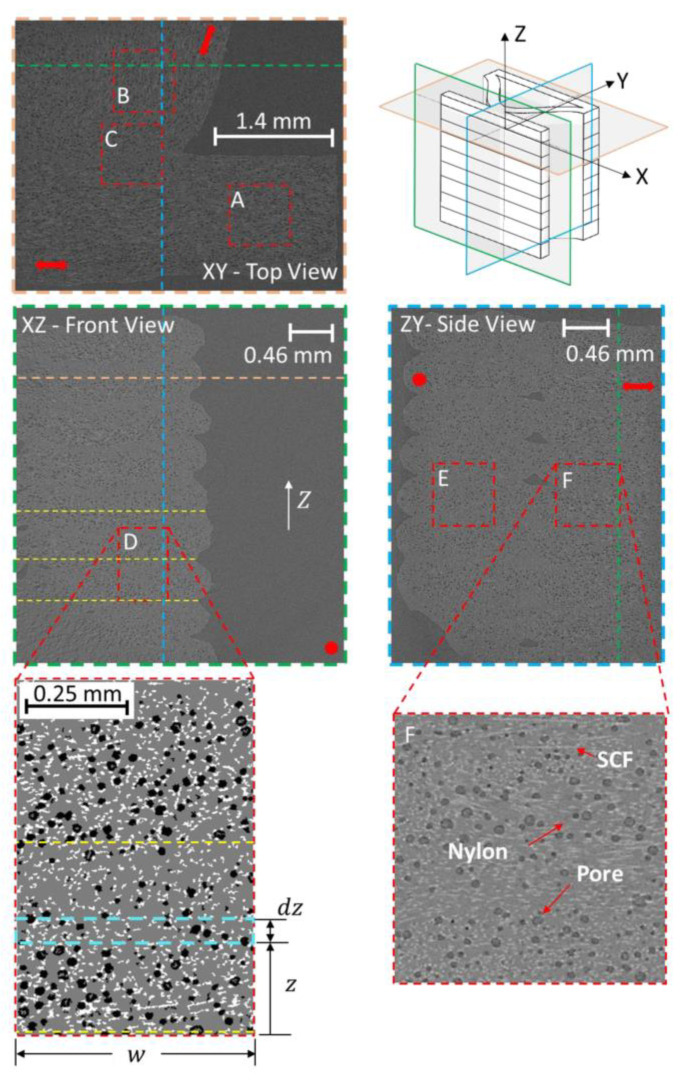
Top, front, and side cross-sectional views and isometric view of CT of MEAM-produced SCF parts. Blue, orange, and green dotted lines and colors represent cross-sectional views of MEX layers and straight and curved raster lines. Red dotted boxes are ROIs (Regions Of Interests) that will be analyzed using a mixed skew-Gaussian distribution (MSGD). The XY view shows the intersection of a straight and curved raster with three ROIs: A—the straight raster, B—the curved raster, and C—the intersection zone. The XZ view shows the MEX layer stacking and the porosity distribution across the layer interface using ROI D with the porosity distribution determined in w x Δz areas along the Z-direction. The ZY view shows the side cross-section of the straight (ROI E) and curved (ROI F) regions [98].

**Figure 19 polymers-14-04941-f019:**
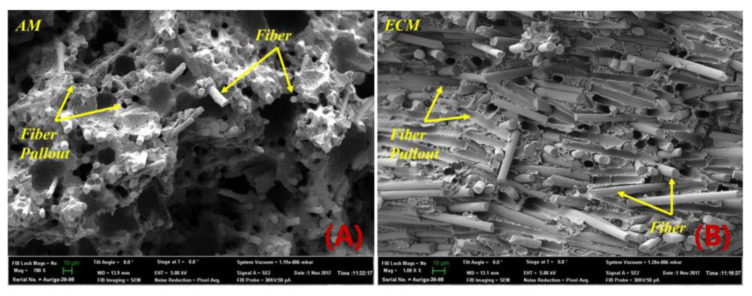
SEM micrographs of a fracture surface of tensile samples; (**A**) MEAM sample: Most of the fibers are aligned in printing direction, also contain voids; (**B**) Compression-molded sample: Fibers are well distributed in all directions [100].

**Figure 20 polymers-14-04941-f020:**
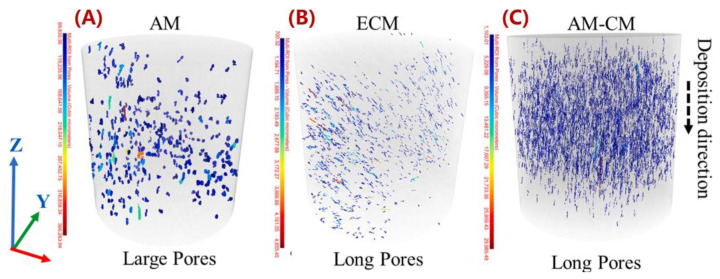
xCT for high aspect ratio pores along the fibers (**A**) AM, (**B**) ECM, and (**C**) AM–CM [101].

**Figure 21 polymers-14-04941-f021:**
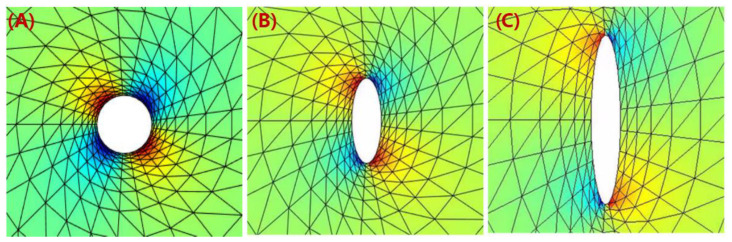
Pressure Distribution around fiber surface: (**A**) $\gamma_{e}=1$; (**B**) $\gamma_{e}=3$; (**C**) $\gamma_{e}=6$. Of note, $\gamma_{e}$ represents the fiber aspect ratio of an ellipsoidal fiber.

**Table 1 polymers-14-04941-t001:** A summary of recent review articles discussing polymer composites in AM techniques.

Authors	Year of Publication	Highlights
Parandoush and Lin [84]	2017	General review for polymer composites in additive manufacturing, including FDM, LOM, SLA, and SLS; Macro mechanical performance and micro-structural characterization on fiber composites produced via each of the techniques are reviewed; The 4D printing of active polymer composites and their functional products are introduced; Modeling approaches for additive manufactured fiber filled polymers, including the short fiber composite theory, classical laminate plate theory, and finite element method-based RVE approach.
Brenken et al. [6]	2018	Specifically focuses on the fiber-filled composites produced by MEAM; The material properties of different composite materials reported by a list of studies are well summarized; Key components of the extrusion deposition process are reviewed, including the material flow and inducing fiber orientation, interlayer bonding mechanism, material solidification, and post-manufacturing deformation and residual stress.
Goh et al. [85]	2019	Novel material developments in polymer composites’ additive manufacturing, including FFF, LDM, SLA, SLS techniques, and so forth; Material performances of AM-fabricated composites are collected, wherein the continuous fiber-reinforced ones exhibited more promising properties as compared to discontinuous fiber-filled composites; The interlayer properties and interface properties between fiber and polymer matrix are stated as a common and profound challenge to be addressed.
Fallon et al. [86]	2019	Highly filled fiber composites in MEAM are focused on; Addressing the concern on how fiber fillers reduces the processability of the feedstock in extrusion/deposition process; The viscosity (e.g., clogging, excessive extrusion force) and material micro-structure of the composites’ feedstock (e.g., interlayer sintering, filament spooling, fiber breakage, fiber orientation, material anisotropy) are the main limitations.
Papon and Haque [82]	2020	Specifically focused on MEAM-produced products; Material performances of a list of continuous or discontinuous fiber-reinforced composites are summarized; Reinforcing fillers in nano/micro/macro scales are discussed. Another interest of this paper is the modeling efforts that address the material behaviors in MEAM.
Daminabo et al. [3]	2020	Mainly focused on MEAM, including the basic setup, meso-structural limitations (e.g., overhang, stringing, warping, structural inhomogeneity), and material feedstock that is regularly employed (e.g, ABS, PLA, etc.); In addition, special feedstock materials (e.g., lignin, cellulose, and nanocellulose) and special functionality of AM products based on these materials are introduced.

**Table 2 polymers-14-04941-t002:** Micro-structural formation studies of MEAM–DFRPCs.

Microstructure	Gained Knowledge	To Be Done
Fiber orientation	Topographic analyses on fiber orientation in the extrusion nozzle [94] and deposited beads [66,91,92,93,98,99]; 2D modeling on the flow-induced fiber orientation in MEAM material flow [63,110,111,112,113,114,115,116,117,118,119,120,121,122].	3D modeling of the MEAM material flow and associated fiber orientation.
Fiber attrition	Statistical study on the fiber length distribution of MEAM–DFRPCs [66,126,127]; Statistical and tomographic analyses on fiber length evaluation through screw-based polymer processing [128,129,130,131,132,133,134,135,136]; Statistical analyses on fiber length distribution of large-scale MEAM-produced composites [67,100,137,138]; Numerical modeling of the fiber length attrition with buckling physics [140,141].	Numerical models specially designed for MEAM processes, and short discontinuous fiber-filled polymers.
Micro-voids	Topographic analyses on the micro-voids within MEAM–DFRPCs [95,96,98,99,100,101]; Numerical flow modeling indicating the pressure distribution around a single fiber attributed to the generation of micro-voids [146].	Systematic modeling of the micro-voids’ formation along the MEAM process.

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
