# Peer review of "A Review on Microstructural Formations of Discontinuous Fiber-Reinforced Polymer Composites Prepared via Material Extrusion Additive Manufacturing: Fiber Orientation, Fiber Attrition, and Micro-Voids Distribution"

_polymers, 2022, doi:10.3390/polym14224941_

Round 1

Reviewer 1 Report

The authors did good work to present the correlations between the MEAM processes and the associated microstructures of the produced composites.

However, there are a few things needed to be handled:

The paper needs updated references related to the impact of the inclusions and voids caused by 3D printing technology, like:

Investigating the Impact of Inclusions on the Behavior of 3D-Printed Composite Sandwich Beams

The impact of the interfacial stresses of the fiber/matrix on the failure is missing.

Fig 3 has to include the dimension range

Fig 4 scale is missing 

Diagram in Fig 5 is not clear

Fig 7 has to include a reference 

Lower row diagrams in Fig 10 are very small

Fig 12 is not clear

The scale in Fig 17 is missing, as well as in Fig 19

Author Response

We greatly appreciate the kind suggestions provided by the reviewer. Please see our attachment for a point-by-point response.

Kind regards,

The authors

Reviewer 2 Report

Comments

In this paper, the authors summarize the current state-of-art in exploring the correlations between the MEAM processes and the associated microstructures of the produced composites. Experimental studies and numerical analyses accompanying with the fiber orientation, fiber attrition, and micro-voids are collected and discussed, respectively. Review content is substantial. However, there are still some issues to be addressed. The specific comments can be found as following:

1.    In keywords, there should be no semicolon after the last keyword. In addition, some of the keywords are too long.

2.    The colors marked with serial numbers in Figure 5 can be changed to colors other than red, yellow, blue and green in the figure, so that the picture can be expressed more clearly.

3.    Molecular orientation is important for the mechanical performance of short fibers as reinforcements. Authors are suggested to add this issue in the revision with some supporting articles on the molecular orientation: Molecular orientation in electrospun fibers: from mats to single fibers; Morphology, polymorphism behavior and molecular orientation of electrospun poly (vinylidene fluoride) fibers; Molecular orientation in aligned electrospun polyimide nanofibers by polarized FT-IR spectroscopy; Temperature-induced Molecular Orientation and Mechanical Properties of Single Electrospun Polyimide Nanofiber.

4.    The marks A) and B) in several figures of the paper are inconsistent with the case format of the notes below.

5.    All the figures should be rechecked and modified to have a better readability, especiall the texts in the figures.

6.    Why is there a vertical line on the left of A) in Figure 16?

7.    In the conclusion, do the authors have some ideas to add about the lack in the numerical studies for further exploring the correlations between the MEAM material flow kinematics and the microstructural formation of the composite beads? In addition, more challenges and possible solutions should be added to guide the future studies.

8.    As a review article, comprehensive literature collection is necessary. Electrospun fibers are important as reinforcements, also including the short electrospun fibers. This topic should also be described as one more sub-section to enrich this review article. Please carefully read and refer the following articles:

1)          Review: A focused review of short electrospun nanofiber preparation techniques for composite reinforcement; Electrospun nanofiber reinforced composites: a review;

2)          Preparation of short electrospun fibers: Short electrospun fibers by UV cutting method; One-step fabrication of short electrospun fibers using an electric spark; ACS Applied Materials & Interfaces 12 (16), 19006-19014, 2020;

3)          Short electrospun fibers as reinforcements: Short nylon-6 nanofiber reinforced transparent and high modulus thermoplastic polymeric composites; Short electrospun polymeric nanofibers reinforced polyimide nanocomposites.

9.    There are still some typos and grammar issues in the manuscript. Authors should carefully recheck the whole manuscript. In addition, full reference information should be added, especially the volume and page numbers.

Author Response

(The authors gave the same response as above.)

Round 2

Reviewer 2 Report

Accept in present form

Author Response

Many thanks for the reviewer's decision of acception. We sincerely appreciate the reviewer's valuable suggestions that significantly improve the quality of our manuscipt. 

With best wishes,

The authors